# Social anxiety influences the stress-buffering potential of social presence: Evidence from cardiovascular and affective reactivity under stress

Antonio Maffei[1,2]*, Elena Scarpa[1,3], Elisabetta Patron[4], Paola Sessa[1,2]

**1** Department of Developmental Psychology and Socialization, University of Padova, Italy, **2** Padova Neuroscience Center, University of Padova, Italy, **3** IRCCS San Camillo Hospital, Italy, **4** Department of General Psychology, University of Padova, Italy

* antonio.maffei@unipd.it

## Abstract

Social presence and social support are fundamental instrumental sources of interpersonal emotional regulation, playing a crucial role in mitigating the impact of stress and negative emotions. This study aims to improve our understanding of the role of relationship type and individual differences in social anxiety in the stress-buffering provided by co-presence of others during stress. A dyadic version of the Trier Social Stress Test (TSST) was employed to experimentally induce stress in a sample of 40 dyads, each comprising a *target* participant who was paired with a second participant, acting as a *supporter*. In half of the dyads the *target* and the *supporter* were in a romantic relationship (Partner Group), while in the other half they were randomly paired (Stranger Group). Heart rate and psychological stress reactivity were collected during the TSST. Results revealed that participants in the Partner Group exhibited a lower heart rate during the acute stress compared to the participants in the Stranger Group, highlighting that the buffering of the physiological stress activity is stronger within close relationships. Nonetheless, participants in the Partner Group reported more anxiety and stress during the task. Furthermore, social anxiety showed a positive correlation with subjective stress reactivity in the participants in the Partner Group, suggesting that it may hinder the perceived benefit of social support. These findings increase our understanding of stress-buffering mechanisms, emphasizing the interplay between social support, stress reactivity, and interpersonal affective processes, also highlighting the need for additional research characterizing individual differences in social regulation of stress.

## Introduction

When overwhelmed by emotions, especially negative ones, people typically seek help and support from others. This natural tendency to rely on others is a

**Data availability statement:** The dataset and analyses reported in this manuscript are available at the Open Science Framework repository: https://osf.io/652b8/?view_only=c6aee289f-aaf49f792854025b60028cd

**Funding:** The author(s) received no specific funding for this work.

**Competing interests:** The authors have declared that no competing interests exist.

fundamental element of emotional behavior that can be observed throughout the lifespan from infancy to old age [1,2]. But why do we behave like this? Modern psychological theories suggest that the core of this behavior is rooted in the social nature of our species, because social support can buffer strong stressful emotions, promoting better coping, thus reducing their burden on both psychological and physiological domains [3–6].

From a psychobiological standpoint, responding to a threat in the environment (*i.e.,* a stressor) is quite a complex phenomenon that begins in the cortex, responsible for sensory perception and cognitive appraisal of the threat, then unfolds primarily along two interacting pathways, the hypothalamus-pituitary axis (HPA) and the autonomous nervous system (ANS) [6,7]. The activity of the HPA axis leads to a cascade release of several neuropeptides (corticotropin-releasing hormone, arginine vasopressin, and adrenocorticotropic hormone) which eventually result in a substantial release in the blood flow of glucocorticoids (GCs) by the adrenal glands. As a consequence of this massive injection of GCs, energy is rapidly mobilized and made available to muscles, the immune system is stimulated, the heart pumps faster, attention becomes more focused, and cognitive efficiency is enhanced [7]. This complex physiological response is constantly regulated through feedback mechanisms involving both central and peripheral districts to keep the stress reaction under control. Indeed, chronic activation of the stress response has been repeatedly shown to have serious detrimental effects on physiological homeostasis and health, in both animal models and humans [7].

This complex psychophysiological response is known to be influenced by the availability of social support, such as the presence of a conspecific in the environment. This influence has been observed and studied primarily in primates, humans included, but research on this topic spans also the psychophysiology of other mammals, as well as, other non-mammalian vertebrates (*i.e.,* birds and fish) [3,5,8,9]. In the context of human psychophysiology, research investigating the interaction between social support and stress reactivity is typically framed within the *stress-buffering hypothesis* [4]. According to this model, social support has the potential to modulate stress reactivity, especially when the stressor has a psychosocial nature. This protective effect translates into a reduced risk of developing the negative sequelae of chronic stress [10], while lack of social support would have a detrimental effect on an individual's ability to deal with stressors, and is a negative predictor of their health status [11].

The empirical support for this theoretical model comes from a large variety of evidence, including epidemiological research [12,13], longitudinal and cross-sectional studies, and, to a lesser extent, experimental laboratory studies [5]. Experimental studies that investigated social buffering of stress reactivity typically adopt paradigms in which a person (*target*) is subjected to a stress task either alone or with a second person (*supporter*) acting as a source of social support. In this paradigm, social support could be either passive, in which case the *supporter* is together with the *target* while they face the stressor but they do not interact, or active, in which case the *supporter* enacts some form of calming/supporting/soothing behavior to actively buffer the stress of the *target*. Studies employing this dyadic paradigm consistently showed

that physiological markers of stress, such as salivary cortisol and cardiovascular reactivity, are reduced when the stressor is faced together with another person compared to when faced alone.

Despite the considerable amount of evidence in support of the *stress-buffering hypothesis*, there are still many open issues about the psychological and psychophysiological mechanisms subtending the stress-buffering potential of social support, which hamper the possibility of efficiently translating these research findings to the clinical setting.

Currently, a significant limitation of this model lies in the unknown factors that can influence the effectiveness of social stress buffering, either by enhancing or diminishing its impact. Among these, one important issue that remains unclear is the role of the relationship type that links the *target* to the *supporter*.

Previous experimental studies suggest that the buffering effects are larger when the *target* who performs a stress task is involved in a close relationship with the *supporter* [14–16]. Nonetheless, these studies usually compared an experimental condition in which the *target* and the *supporter* are related to each other (*e.g.,* romantic partners or friends) with a control condition in which the *target* performs the stress task alone. As a consequence, this does not allow disentangling the general contribution of the mere presence of others in times of trouble, from the specific contribution provided by the presence of a significant other when enduring emotional stress.

A second issue that needs a better understanding is represented by how stress-buffering is reflected in the psychological domain, that is if the reduction of physiological stress reactivity parallels a reduction in the levels of experienced stress and anxiety. Intuitively, we would expect that benefiting from social support during stress would reduce the experience of stress-induced negative emotions, but the evidence in this direction is little, with several authors reporting that the presence of a partner during a stressful task, compared with a control condition in which participants were alone, does not prompt a reduction in the levels of negative emotions experienced [14,17,18]. In their review, Uchino and colleagues [18] discussed potential explanations for this lack of evidence, suggesting exploring in more detail the social context acting as a source of social support. Specifically, they suggest that, in times of acute stress, close persons forming the primary social ties might be not efficient in providing support because they do not have direct experience with the stressor. This would then impact the perception of social support (i.e., *perceived support*), which has been recognized to be the most important predictor of social stress-buffering [10].

Finally, a third issue that urges a better understanding is how the mechanisms of social stress buffering are shaped by individual differences in how people perceive and react to emotional stimuli and regulate their emotions. Specifically, an element of particular interest is the role of dispositional social anxiety. Social anxiety can be defined as the tendency of individuals to be wary of others and to experience discomfort in social interactions. Low levels of social anxiety are normally experienced in social interactions, especially when new and/or unexpected [19]. In a minority of the population affected by a social anxiety disorder, this experience of negative emotions is far more intense and accompanied by strong physiological arousal and leads to a significant impact on their daily functioning and well-being [20]. Indeed, individuals with social anxiety might have trouble accessing and benefiting from social support in times of emotional stress, due to their chronic fear of negative evaluation. Furthermore, these individuals have a biased perception of others, thus they might perceive social support as inadequate or unsatisfactory, even when it is available [21]. Finally, high levels of social anxiety can lead to serious difficulties in forming and maintaining social ties, which in turn can limit the quantity and quality of social support available [22]. In sum, social anxiety can impact both the objective facet of social support and, more importantly, the subjective, or perceived facet of social support, which is unambiguously considered to be the most predictive element of a successful stress-buffering effect [10]. Finally, it is unknown how this facet of individual affective style would interact with the social context (close relationship vs stranger individual) in moderating the social stress-buffering effect.

## Research objectives

In this research we aimed to contribute to clarifying these open issues, thus we designed an experiment in which we measured psychophysiological reactivity in a sample of participants undergoing a stress-induction procedure while

manipulating their social context. Stress was elicited with the public speaking task of the Trier Social Stress Test (TSST), which represents the gold standard for experimental induction of a robust physiological stress response under laboratory conditions [23,24]. Briefly, this task consists of preparing and delivering a mock job interview in front of an evaluative panel. Despite its simplicity, this task proved its efficacy in prompting a strong modulation in many psychophysiological parameters linked with stress reactivity, producing an increase in both cardiovascular (*i.e.,* heart rate, heart rate variability, blood pressure) and hormonal (*i.e.,* cortisol) responses. In this study, one participant was the *target* of the TSST, while the other acted as the *supporter* participant. The *target* was the one undergoing stress, while the *supporter* was simply asked to be present without explicitly interacting with the target, thus acting as a source of *passive social support*. Stress reactivity of the *target* was assessed by continuously measuring their cardiovascular activity using electrocardiography (EKG) and measuring their levels of perceived stress and anxiety throughout the task with self-reports. Additionally, we collected their trait levels of social anxiety.

For what concerns the role played by different types of relationships (*close relationship* vs *stranger individual*), we recruited a group of participants who were in a stable romantic relationship and a group of strangers, predicting that the physiological stress-buffering effects of social support would be more robust within close relationships.

With regards to the psychological domain, we instead predicted to either observe similar levels of negative affect in the two groups, replicating previous findings, or to observe increased negative affect in the Partner group, which would suggest that in an evaluative context, the presence of a close person exacerbates the perception of stress. Finally, we also wanted to explore the role played by individual differences in social anxiety. In this regard, we could expect a positive relationship between social anxiety and stress reactivity in those performing the task with a stranger, but not in those performing the task with a partner. This would suggest that the effects of stress buffering of partners are robust to the individual tendency to experience negative emotions in social contexts. Alternatively, we could observe a positive relationship between social anxiety and stress reactivity in the group of participants performing the task when their partner is present, but not in those performing it when a stranger is present, thus revealing that social anxiety has the potential to diminish the effectiveness of the stress-buffering provided by the presence of a supportive other.

## Methods

### Participants

Eighty healthy young adults (Mean age = 23.3 y, S.D. = 1.9 y) divided into forty dyads were recruited to participate in the present research in exchange for course credits. They were divided according to their relationship status into two groups. One group of dyads comprised participants who were involved in a romantic relationship with each other for at least six months, thus they were assigned to the Partner Group (PG, n = 19), while the other group comprised participants who were randomly matched and assigned to the Stranger Group (SG, n = 21). For each dyad in both groups, one participant (always female) was assigned the role of the *target* and performed the stress task, while the other participant was assigned the role of the *supporter*.

The sample size was estimated *a priori* using G*Power, which suggested a minimum sample of 36 participants for detecting a small-to-medium effect size (*f* = 0.2) in the within-between interaction with a power of 80%. The two groups did not differ in terms of age ($t_{(26)}$ = 1.9, p = 0.06). The experimental protocol was approved by the University of Padova ethics committee for psychological studies (protocol n° 4716), and data were collected between May 15th and July 30th 2022. All the procedures were carried out according to the principles expressed in the Declaration of Helsinki for human research.

### Psychosocial stress induction task

Psychosocial stress in the *target* participants was elicited by the public speaking task of the Trier Social Stress Test (TSST), which consists of four phases: (i) baseline, (ii) stress anticipation, (iii) acute stress, and (iv) recovery, with each phase lasting 5 minutes. The psychosocial stressor consisted of first preparing and then delivering a mock job interview in

front of a mock interviewing panel. To further increase the evaluative stress, participants were told that the interview would be videotaped for further analysis by public speaking experts.

Participants were first acquainted with the laboratory environment for 15 minutes before the start of the experiment, and seated comfortably during the whole task to avoid confounding orthostatic effects on the cardiovascular reactivity.

## Cardiovascular measures of stress reactivity

The cardiovascular activity of the *target* participants was continuously collected throughout the task using three ECG sensors placed on the participant's chest in a modified lead II configuration. The ECG signal was collected and amplified through a BrainAmp amplifier (BrainProducts, Munich, Germany) with a sampling rate of 250 Hz. The raw ECG was pre-processed using the software Kubios (v2.2) to extract the interbeat interval time series with the Pan-Tompkins algorithm, and correct rare artifacts due to participants' movement through local log-likelihood point-process statistics devised for cardiac dynamics. Finally, heart rate (in beats per minute) was computed for each phase of the task.

## Psychological measures of stress reactivity

Task-induced changes in participants' emotional state were collected during each phase of the task, using the state form of the State-Trait Anxiety Inventory (STAI-Y1). The state form of the STAI is a widely used psychometric self-report instrument consisting of 20 items in which participants express how they feel at the moment using a 4-point Likert scale, measuring state anxiety levels. Participants were also asked to express their levels of perceived stress using a 10-point visual-analog scale, ranging from 0 (fully relaxed) to 10 (extremely stressed).

Trait disposition in social anxiety was measured using the Social Interaction Anxiety Scale (SIAS). It is a widely used psychometric self-report instrument consisting of 20 items, in which the participant is asked to express how much fear or anxiety they feel about different social evaluative scenarios, using a 5-point Likert scale.

## Procedure

Once participants arrived, they were informed about the study procedures and were asked to provide their signed informed consent to take part in the research. Both members of the dyad provided their consent. Then, they were led to a dimly lit sound-attenuated room and asked to sit next to each other as shown in Fig 1, and the experimenters attached the ECG sensors to the *target* participant's chest to record their cardiovascular activity.

The task began with a 5-minute rest period for the baseline, in which participants were simply asked to relax and not interact verbally with each other. In the second phase, the *target* participant was asked to mentally prepare a 5-minute speech describing why she would be the ideal candidate for her dream job. Furthermore, she was informed that she would be video-recorded by a webcam (visible to the participant) and that this recording would be later analyzed by public speaking experts. In the third phase, the *target* participant was asked to deliver her 5-minute speech in front of an interviewing panel comprising two experimenters' confederates. During this phase, the *supporter* was asked not to do or say anything (following a passive support protocol), while the experimenters merely encouraged the participant to continue speaking if she paused for more than 20 seconds continuously. Finally, in the fourth phase, the participant was asked to rest for another five minutes. After each phase, the *target* participant was prompted to complete a self-report questionnaire to collect the task-induced changes in her anxiety and perceived stress level. At the end of the experiment, participants were informed about the deceptive nature of the task and that the videotaping did not occur, and debriefed about the study aims.

## Statistical analysis

Linear mixed-effects models were used to analyze the three dependent variables considered in this study (HR, STAI, and perceived stress). For each model, we included as fixed predictors the independent variables Time (4 levels: baseline, anticipation, speech, recovery), Group (2 levels: partner, stranger), and their interaction. The contrasts were set to compare the

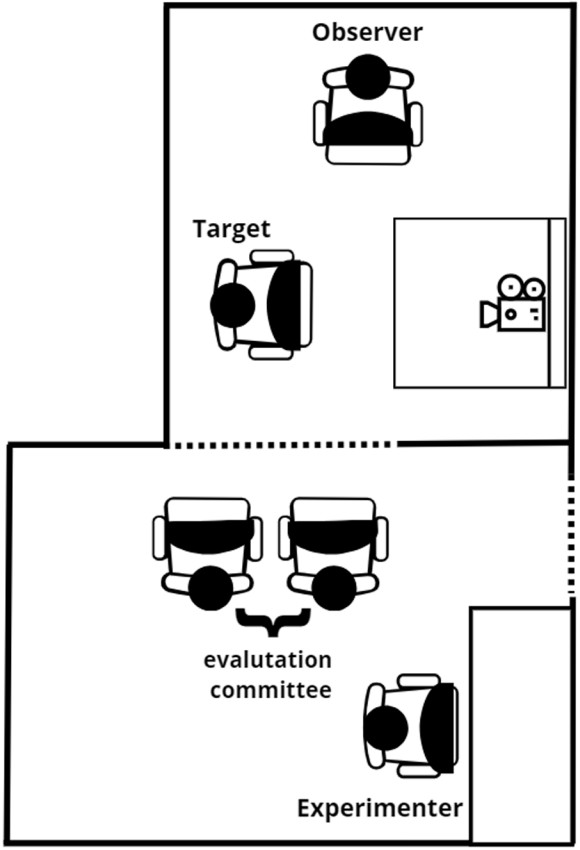

**Fig 1. Schematic depiction of the experimental setting.**

two groups at each level of the Time factor. The non-independence of the observations was modeled by including the random effect of the participant. An F-test with Satterthwaite correction for degrees of freedom was used to test the significance of the fixed predictors, and the size of the effects was quantified with Cohen's $f$. Collinearity of the predictors was evaluated using the variance inflation factor. Finally, the positive influence of dispositional social anxiety on psychophysiological stress reactivity in the two groups was quantified using Pearson's correlation between the SIAS scores and HR, STAI, and perceived stress during the speech phase. All the analyses were carried out using the R software (version 4.1.2).

## Results

### Cardiovascular stress reactivity

The analysis of the heart rate of the *target* participants revealed a significant main effect of Time ($F_{(3,108)}$ = 60.18, $p < 0.001$, $f = 1.26$, VIF = 1) and the predicted significant interaction Time x Group ($F_{(3,108)}$ = 3.14, $p = 0.02$, $f = 0.29$, VIF = 1.22). The analysis of the regression parameters (Table 1) revealed that the heart rate of the participants in the Partner group was lower than the heart rate of the participants in the Stranger group specifically during the acute stress phase ($\beta = 9.15$, $t = 2.817$, $p = 0.005$, $M_{Partner} = 94.2$ bpm, $M_{Stranger} = 103.3$ bpm). The two groups did not differ instead during the baseline ($\beta = -0.62$, $t = -0.132$, $p = 0.89$, $M_{Partner} = 81.6$ bpm, $M_{Stranger} = 81$ bpm), the anticipation ($\beta = 6.05$, $t = 1.879$, $p = 0.06$, $M_{Partner} = 87.9$ bpm, $M_{Stranger} = 93.4$ bpm), or the recovery ($\beta = 2.17$, $t = 0.673$, $p = 0.5$, $M_{Partner} = 78.4$ bpm, $M_{Stranger} = 80$ bpm). These results are shown in Fig 2.

**Table 1. Summary of the model for heart rate data (adjusted R2 = 0.65).**

| Fixed Effect | β | Std. Error | t | p | |
|---|---|---|---|---|---|
| Partner – Baseline (Intercept) | 81.27 | 2.33 | 34.8 | < 0.001 | *** |
| Partner – Anticipation | 9.38 | 1.61 | 5.82 | < 0.001 | *** |
| Partner – Speech | 17.19 | 1.62 | 10.58 | < 0.001 | *** |
| Partner – Recovery | −2.09 | 1.61 | −1.29 | 0.19 | |
| Partner vs Stranger Baseline | −0.61 | 4.67 | −0.13 | 0.89 | |
| Partner vs Stranger Preparation | 6.05 | 3.22 | 1.87 | 0.06 | |
| Partner vs Stranger Speech | 9.15 | 3.24 | 2.81 | 0.005 | *** |
| Partner vs Stranger Recovery | 2.16 | 3.22 | 0.67 | 0.5 | |

## Subjective stress reactivity

Reliability of the STAI was evaluated with Cronbach's alpha, which was 0.94 indicating excellent reliability of the instrument. The two groups did not differ in terms of baseline levels of state anxiety ($t_{(37)}$ = 0.35, p = 0.72). The analysis of the self-reported anxiety of the *target* participants revealed a significant main effect of Time ($F_{(3,108)}$ = 19.85, p < 0.001, $f$ = 0.73, VIF = 1) and a significant interaction Time x Group ($F_{(3,108)}$ = 5.58, p = 0.001, $f$ = 0.38, VIF = 1.05). The analysis of the regression parameters (Table 2) revealed that, contrary to our prediction, the participants in the Partner group experienced greater anxiety than the participants in the Stranger group, specifically during the acute stress phase (β = −12.85, t = −3.95, p < 0.001, $M_{Partner}$ = 57.1, $M_{Stranger}$ = 45.4). The two groups did not differ during the baseline (β = 1.193, t = 0.32, p = 0.74, $M_{Partner}$ = 43.5, $M_{Stranger}$ = 44.7), the anticipation (β = −3.39, t = −1.049, p = 0.29, $M_{Partner}$ = 53.6, $M_{Stranger}$ = 51.4), or the recovery (β = −5.92, t = −1.832, p = 0.07, $M_{Partner}$ = 44.6, $M_{Stranger}$ = 39.9). These results are shown in Fig 3A.

A similar effect was also found for the subjective stress experienced by the participants. The analysis revealed a significant main effect of Time ($F_{(3,112)}$ = 24.5, p < 0.001, $f$ = 0.81, VIF = 1) and a significant interaction Time x Group ($F_{(3,112)}$ = 3.36, p = 0.02, $f$ = 0.3, VIF = 1.1). The analysis of the regression parameters (Table 3) revealed that the participants in the Partner group experienced greater stress than the participants in the Stranger group, specifically during the acute stress phase (β = −2.1, t = −2.792, p = 0.006, $M_{Partner}$ = 7.9, $M_{Stranger}$ = 6.1). The two groups did not differ during the baseline (β = 0.26,

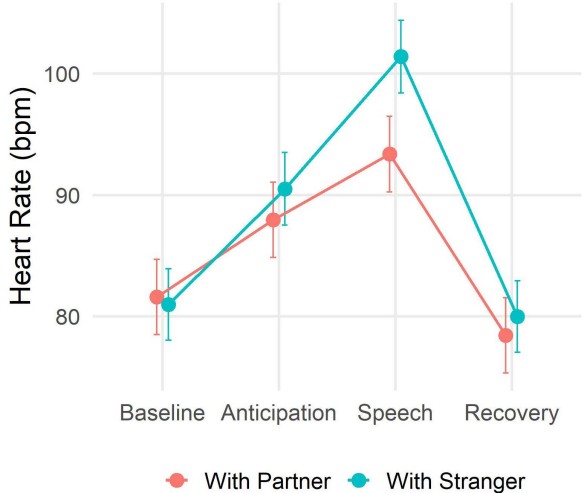

**Fig 2. Heart rate for the two groups in each phase of the TSST.** Bars represent standard errors.

**Table 2. Summary of the model for self-reported anxiety (adjusted R2 = 0.48).**

| Fixed Effect | β | Std. Error | t | p | |
|---|---|---|---|---|---|
| Partner – Baseline (Intercept) | 44.07 | 1.83 | 24.07 | < 0.001 | *** |
| Partner – Anticipation | 8.41 | 1.61 | 5.2 | < 0.001 | *** |
| Partner – Speech | 7.2 | 1.62 | 4.42 | < 0.001 | *** |
| Partner – Recovery | −1.8 | 1.61 | −1.11 | 0.26 | |
| Partner vs Stranger Baseline | 1.19 | 3.66 | 0.32 | 0.74 | |
| Partner vs Stranger Preparation | −3.39 | 3.23 | −1.04 | 0.29 | |
| Partner vs Stranger Speech | −12.85 | 3.25 | −3.94 | < 0.001 | *** |
| Partner vs Stranger Recovery | −5.92 | 3.23 | −1.83 | 0.06 | |

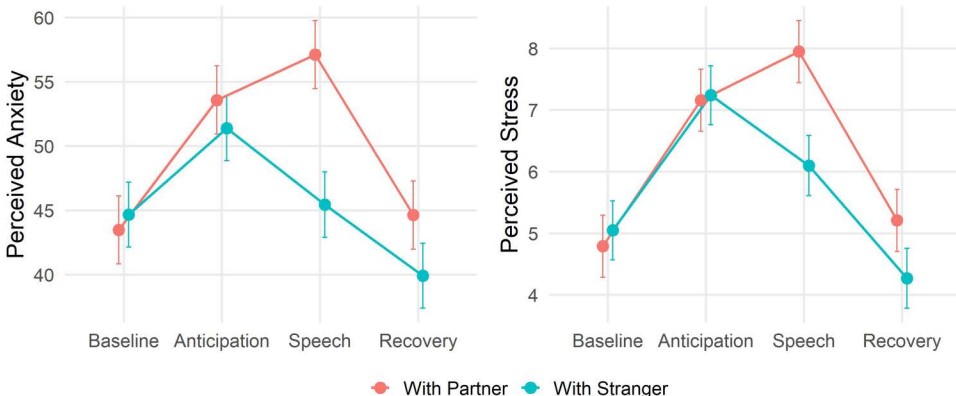

**Fig 3. The left panel shows the levels of perceived anxiety for the two groups in each phase of the TSST.** The right panel shows the levels of perceived stress for the two groups in each phase of the TSST. Bars represent standard errors.

t = 0.37, p = 0.71, $M_{Partner}$ = 4.79, $M_{Stranger}$ = 5.05), the anticipation (β = −0.18, t = −0.23, p = 0.81, $M_{Partner}$ = 7.16, $M_{Stranger}$ = 7.24), or the recovery (β = −1.19, t = −1.59, p = 0.11, $M_{Partner}$ = 5.21, $M_{Stranger}$ = 4.27). These results are shown in Fig 3B.

## Association between trait social anxiety and acute stress reactivity

Reliability of the SIAS was evaluated with Cronbach's alpha, which was 0.91 indicating excellent reliability of the instrument. The two groups did not differ in terms of their levels of social anxiety ($t_{(37)}$ = 0.53, p = 0.59). The correlation analysis between acute cardiovascular stress reactivity with dispositional levels of social anxiety did not reveal any association between the two neither in the participants in the Partner group (r = 0.19, p = 0.22) nor in the participants in the Stranger group (r = 0.19, p = 0.23). On the other hand, we found a significant association between subjective stress reactivity and dispositional levels of social anxiety. Specifically, we found in the Partner group a positive association between the levels of anxiety experienced by these participants during the speech phase and their levels of trait social anxiety (r = 0.41, p = 0.03). This association was absent (r = 0.03, p = 0.44) in participants who performed the task in the presence of a stranger. Similarly, we found a positive association between social anxiety and perceived stress in the Partner group (r = 0.48, p = 0.01), but not in the Stranger group (r = −0.21, p = 0.81). These results are shown in Fig 4.

To exclude that the observed pattern could be explained by a simpler difference in the levels of social anxiety between the two groups, we compared the SIAS scores with a t-test, which showed that both groups had comparable levels of trait social anxiety ($t_{(38)}$ = 0.53, p = 0.59, $M_{Partner}$ = 32.5, $M_{Stranger}$ = 30.1).

**Table 3. Summary of the model for self-reported stress (adjusted R2 = 0.39).**

| Fixed Effect | β | Std. Error | t | p | |
|---|---|---|---|---|---|
| Partner – Baseline (Intercept) | 4.91 | 0.34 | 14.17 | < 0.001 | *** |
| Partner – Anticipation | 2.27 | 0.37 | 6.08 | < 0.001 | *** |
| Partner – Speech | 2.1 | 0.37 | 5.57 | < 0.001 | *** |
| Partner – Recovery | −0.17 | 0.37 | −0.47 | 0.63 | |
| Partner vs Stranger Baseline | 0.25 | 0.69 | 0.37 | 0.71 | |
| Partner vs Stranger Preparation | −0.17 | 0.74 | −0.23 | 0.81 | |
| Partner vs Stranger Speech | −2.1 | 0.75 | −2.79 | 0.006 | *** |
| Partner vs Stranger Recovery | −1.19 | 0.75 | −1.58 | 0.11 | |

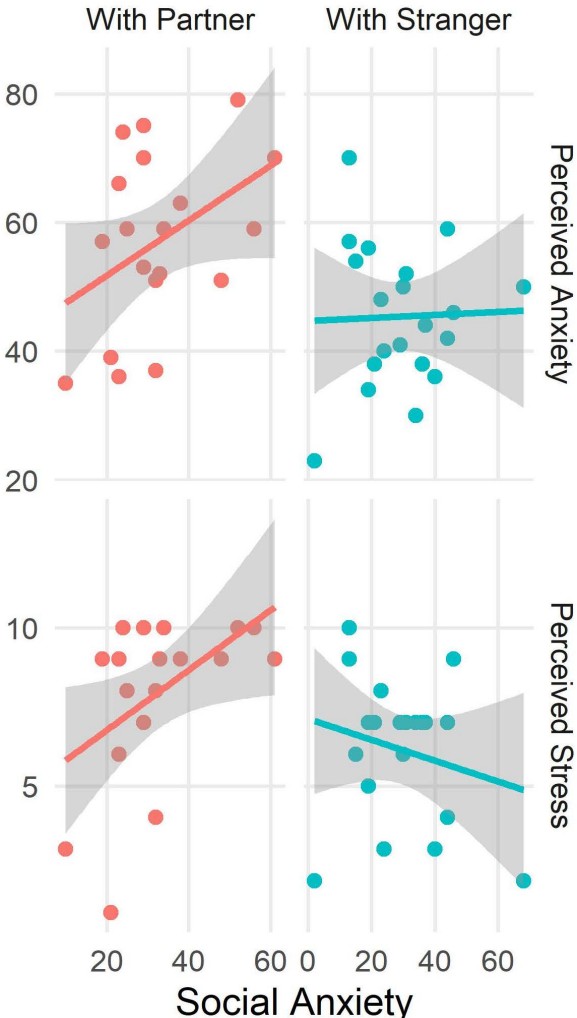

**Fig 4. The top panel shows the correlation between social anxiety and perceived anxiety during the speech phase for the two groups.** The bottom panel shows the correlation between social anxiety and perceived stress during the speech phase for the two groups.

## Discussion

Social bonding is an essential source of emotional regulation for humans, especially during moments of strong negative emotions. The *stress-buffering hypothesis* posits that the perception of available social support in times of emotional stress buffers the sharp psychophysiological *fight-or-flight* stress response, leading to a series of beneficial outcomes on the individual psychological and physical health [4,10].

The most important of these outcomes is that social stress buffering exerts a protective effect on the cardiovascular health of individuals who can benefit from an established network of supportive social relationships [5,10,13]. This has been proved in several epidemiological studies, however, there are still many open issues regarding the mechanisms by which stress buffering operates and regarding what are the best predictors of its success. This experimental study aimed to contribute new knowledge able to clarify if the stress-buffering effect is amplified in the presence of close social support, and how it is affected by individual differences.

Employing a modified version of the TSST, adapted to accommodate two participants, one as the *target* of the stressor and another as the *supporter*, and manipulating the relationship status between the *target* and the *supporter* we first sought to understand if the stress-buffering effects of co-presence is stronger within established relationships, or is it independent of the relationship type linking the *target* and the *supporter*.

Comparing the degree of cardiovascular activation during the task, we observed that participants who were facing the stress together with their partner showed a lower peak of heart rate during the acute stress phase, compared to participants performing the task together with a stranger.

This result suggests that close social support is critical in mitigating the degree of physiological stress response. This extends previous reports showing that when the *target* and the *supporter* share an affective relationship (*e.g.,* romantic partners or friends) the physiological effects of the stress-buffering are larger, compared to a condition in which the *target* faces the stressor alone [14,16]. Indeed, in this research, we provide experimental evidence that the strength of stress buffering on the cardiovascular system crucially depends on the type of affective relationships building up a dyad. An analogous psychophysiological pattern has been already discovered in the developmental context, showing that psychophysiological emotional reactivity in children is lower when their mother is present, highlighting the crucial role of the maternal buffering of children's stress response [1,6]. Similar patterns have also been observed in non-human primates [25]. Here we extend this evidence to the adult population, supporting the idea that, at least in terms of psychophysiological activation, the individual's social network can be a strong instrumental source of emotional co-regulation.

Concerning the potential mechanisms supporting this effect, our result should be read in the light of the Social Baseline Theory's main predictions [26,27], to which our study provides additional and independent support. The SBT model posits that humans are inherently adapted to a *social* ecological niche, thus our neurocognitive mechanisms are optimized to operate in a social context [26]. When it comes to affective behavior and its psychophysiological underpinnings, SBT would predict that the degree of activation in response to a stressor faced alone would be larger compared to the degree of activation when facing a stressor together with someone else [28,29]. Here, we add to this knowledge by showing that a close social context, such as a romantic partner, prevents the individual under stress from investing additional physiological effort into an allostatic upregulation of their defensive system, eventually resulting in a lower level of cardiovascular stress.

For what concerns the analysis of the participants' self-reported emotional state, we observed that participants performing the task together with a supportive partner felt more anxious and stressed, compared to participants paired with a random stranger. This result provides additional support to the growing idea that stress-buffering does not operate with either the same strength or in the same direction in the psychological and physiological domains [17,18]. Furthermore, it reveals that in terms of psychological experience, the presence of a close person might even amplify, rather than reduce, negative emotions experienced in an evaluative context. Indeed, it is important to consider that the task employed in this research, the TSST, is characterized by a marked socio-evaluative component which could have very likely triggered in

the participants a strong emotional concern outweighing the potential benefit of having their partner present acting as a *safety signal* [18,30]. Indeed, it has been shown that attention to safety cues is reduced when it is not possible to exert control over an emotionally-charged situation [31–33]. Thus, the observed pattern could reflect that participants performing in front of their partners felt an additional evaluative pressure, a pressure that those performing in front of a stranger did not experience.

The correlation analysis performed between the levels of negative emotions experienced during the acute stress phase and trait social anxiety provides support for this interpretation. The results of this analysis showed that, for the individuals facing the stressor with their partners, these two dimensions are strongly associated. Thus, the tendency to experience distress in evaluative situations amplifies the strength of negative emotions in participants paired with their partners, while it plays no role in the levels of stress experienced by participants doing the task with a stranger.

Taken together, these results suggest that social anxiety might act as a hidden influence on the perceived component of social support, thus explaining why individuals with a high fear of social evaluation struggle to benefit from the presence of a supporting other when facing a stressor. Specifically, participants with high levels of social anxiety might lack the cognitive resources to implement a reappraisal of the threat in light of the available social support. This is because their resources are already budgeted for regulating their fear of making a bad impression in front of their partner [34]. Participants who were paired with a stranger, instead, did not have to regulate their emotions for this additional element, leading to the counterintuitive effect of them reporting lower levels of stress and anxiety.

Interestingly, we did not observe any moderating role of social anxiety on the differential pattern of cardiovascular reactivity in the two groups.

This clear dissociation would suggest that the mechanisms of social stress buffering might be characterized by different levels of specificity in the two domains, affective and psychophysiological. Our results would cautiously support the idea that lower psychophysiological stress response in a supportive social context is a general-domain mechanism, resilient to the individual differences which play, instead, a crucial role in moderating how the stress-buffering effect is consciously appraised at the emotional level.

Furthermore, the dissociation between the psychological and physiological domains would also advise for a more comprehensive account of the social stress buffering phenomenon and, more generally, the social regulation of emotions [29,35,36]. Indeed, the results of this research highlight that these phenomena are more complex than suggested by current models. This requires the need to deepen further our understanding of how individual differences in affective style could amplify or impair the effects of social relationships on the individual's ability to deal efficiently with negative emotions.

## Conclusions and limitations of the study

In conclusion, it is important to highlight that our results should not be interpreted without considering the limitations of this research. First, our sample comprised only female participants in the role of the *target*. This choice was made to minimize the potential confounding effects of the gender differences that exist in affective and psychophysiological reactivity [37–39], and in line with previous research showing that stress buffering on autonomic reactivity is larger in women [5]. Nonetheless, this warns against a generalization of these effects also to the male population. A second limitation of this research is that our sample comprised only young adults, who were engaged in affective relationships lasting between one and two years and were not living together. This could limit the generalization of these results to older adults who are typically involved in longer affective relationships and often live together, thus they could not display the very same pattern of psychophysiological co-regulation under stress [40]. A third limitation is that in our study we focused on the simple co-presence of another person as a proxy for social support. We acknowledge that in everyday life social support is a complex construct that spans well beyond simple co-presence [10]. On the other hand, the aim of this work was to improve the experimental evidence in favor of the social stress buffering dynamics. Thus, we limited our manipulation

to co-presence in order to minimize the inherent variability that could arise from asking the supporters to provide active support to the targets under stress. Future investigation should focus on designing more complex manipulation of social support. Indeed, a potential alternative interpretation for finding that participants performing in front of their partner report higher anxiety than those performing in front of a stranger, is that the former could have felt more comfortable in disclosing their levels of negative emotions due to their partner's presence. Explicitly manipulating the behavior of the partner could help clarify when its presence can act as a source of emotional buffer or as an additional source of psychological distress. A fourth limitation is represented by the fact that only half of the participants included in the Stranger condition were in a romantic relationship at the time of the research. At present, we can not rule out that being in a romantic relationship might have an impact even when facing a stressor together with a stranger providing support. In future studies, this feature should be controlled in order to evaluate possible differences in stress reactivity that could depend entirely on being or not in a romantic relationship, independent from having support when facing a stressor. A fifth and final limitation is that we decided to focus on social anxiety as one of the many facets of individual differences moderating the process of stress buffering and social regulation of emotions. We do expect that future research will investigate further the role of affective style, especially with regards to the role of individual differences in emotion regulation abilities [41,42] aiming at uncovering how they contribute to amplifying or impairing the process of stress buffering.

## Author contributions

**Conceptualization:** Antonio Maffei, Elisabetta Patron, Paola Sessa.

**Data curation:** Antonio Maffei.

**Formal analysis:** Antonio Maffei.

**Investigation:** Elena Scarpa.

**Methodology:** Antonio Maffei, Elena Scarpa, Elisabetta Patron, Paola Sessa.

**Project administration:** Antonio Maffei.

**Supervision:** Paola Sessa.

**Visualization:** Antonio Maffei.

**Writing – original draft:** Antonio Maffei, Elena Scarpa.

**Writing – review & editing:** Elisabetta Patron, Paola Sessa.

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
