## [Decision Letter · Decision Letter 0]

11 Feb 2025

PONE-D-24-26805

Social anxiety influences the stress-buffering potential of social presence: Evidence from cardiovascular and affective reactivity under stress

PLOS ONE

Dear Dr. Maffei,

Our referees have now considered your paper and have provided good feedback. The comments of the reviewers are included at the bottom of this letter. The reviewers find your submission worthwhile but suggest minor revisions to improve your manuscript before it can be considered for publication. This assessment aligns with my own evaluation of your submission. Therefore, I invite you to revise your manuscript. Upon re-submission, your article will undergo another round of review by the same referees to assess the revisions.

Please make sure that you address or at least respond to all reviewers' comments.

We look forward to receiving your revised manuscript.

Kind regards,

Guangyu Zeng, Ph.D.

Academic Editor

PLOS ONE

Journal Requirements:

Reviewers' comments:

Reviewer's Responses to Questions

**Comments to the Author**

1. Is the manuscript technically sound, and do the data support the conclusions?

Reviewer #1: Yes

Reviewer #2: Yes

2. Has the statistical analysis been performed appropriately and rigorously? 

Reviewer #1: Yes

Reviewer #2: No

3. Have the authors made all data underlying the findings in their manuscript fully available?

Reviewer #1: Yes

Reviewer #2: Yes

4. Is the manuscript presented in an intelligible fashion and written in standard English?

Reviewer #1: Yes

Reviewer #2: No

5. Review Comments to the Author

Reviewer #1: The current study tested whether the presence of a romantic partner buffered women from the stress of experiencing the Trier Social Stress test and whether social anxiety moderated this effect. The topic is important and the paper well-written. I have a few comments and concerns, listed below:

Introduction:

- There were spots throughout the introduction where a citation was needed and there isn’t one. For example, the definition of social anxiety. Please add more citations throughout the introduction to back up claims.

- Was there a directional hypothesis about the relation between social anxiety and stress reactivity in each group. Different possibilities were discussed but there did not appear to be a specific hypothesis.

Methods:

- Were those in the stranger condition in romantic relationships? I understand that they were paired with strangers but did they have relationships outside of this study? If not, there might be systematic differences between the groups. There might be differences in how people perceive support from others based on their own relationship experiences so this might be listed as a limitation if they were not in relationships.

- Was there rationale for always having the target participant be female? This was not set up in the introduction and came as a surprise.

- What was the reliability of the STAI and SIAS scales in the current sample?

- Did the two groups differ on any demographic variables? Baseline statistics should also be reported, perhaps in a table.

Results:

- Were there any differences in stress reactivity or anxiety based on the gender of the supporter?

Discussion:

- I am struck by the difference in the physiological and psychological results. I agree with the authors’ proposal that the emotional concern might outweigh the safety signal of their partner or that performing in front of a partner is an added evaluative pressure. I have a potential third option as well. The TSST is a fairly negative experience and I’m wondering if it is possible that the target participant feels safer to express negative feelings in the presence of their significant other.

Reviewer #2: Review of the Manuscript ID PONE-D-24-26805 Title: “ Social anxiety influences the stress-buffering potential of social presence: Evidence from cardiovascular and affective reactivity under stress” for the Plos One Journal.

General Comments

From my point of view, it is a very interesting topic and simultaneously it seems that to the best of my knowledge is an empirical research aims to improve our understanding of the role of relationship type and individual differences in social anxiety on the stress-buffering provided by co-presence of others during stress. A dyadic version of the Trier Social Stress Test (TSST) was employed to experimentally induce stress in a sample of 40 dyads, each comprising a target participant who was paired with a second participant, acting as a supporter. In half of the dyads the target and the supporter were in a romantic relationship (Partner Group), while in the other half they were randomly paired (Stranger Group). Heart rate and psychological stress reactivity were collected during the TSST. Results revealed that participants in the Partner Group exhibited a lower heart rate during the acute stress compared to the participants in the Stranger Group, highlighting that the buffering of the physiological stress activity is stronger within close relationships. Nonetheless, participants in the Partner Group reported more anxiety and stress during the task. Furthermore, social anxiety showed a positive correlation with subjective stress reactivity in the participants in the Partner Group, suggesting that it may hinder the perceived benefit of social support. These findings increase our understanding of stress-buffering mechanisms, emphasizing the interplay between social support, stress reactivity, and interpersonal affective processes, also highlighting the need for additional research characterizing individual differences in social regulation of stress.

The paper consists of following sections: Introduction, Methods, Results, Discussion, Conclusion and Policy Implications.

However, I find some recommendations:

1. The Manuscript needs careful English proofreading because there are some shortcomings. For instance, the article “the” is sometimes missing in front of nouns, the message in some paragraphs is not clear enough. It looks like the first part was written by one author with a greater command of the English language, and the rest of the paper was written by someone else. The numerous grammar errors made this a difficult paper to read. It was strange to see the authors refer to tables that were not submitted. I was unable to find any supplementary material to the submission, so I think this was truly omitted by the authors. Please read the manuscript carefully.

2. It would be very useful to add in the "Introduction" section the purpose, objectives and hypothesis of the research. I consider that a weak point of the paper is that the authors did not show the novelty of the paper compared to other works. That is why, I consider that the introduction should specify the novelty of the paper compared to other papers published in this area.

3. The research is well based on science and the results are in agreement with the theoretical part. From my point of view, the paper is original and the topic addressed brings added value to the specialized literature regarding the anxiety. The paper is well written and easy to read.

4. The research is well based on science and the results are in agreement with the theoretical part. The model applied to the analyzed data is correctly used in the analysis undertaken, it is a strength point of this paper.

5. The Literature Review and the conclusions sections cannot be misssing from the paper.

6. In my opinion, the results from the Statistical Analysis section, which presents three models with dependent variables and random effects, should be presented in a table. Here, the authors should apply the Hausman and VIF tests (see: https://doi.org/10.1371/journal.pone.0283277).

7. Authors must refer to the software used for econometric analyses: STATA, EViews, SPSS etc.

8. The authors must also show the values of the adjusted R-square, which is more relevant in the methods used in this paper.

9. The authors applyed the regression method, then they must apply the following tests: the authors must to apply an econometric method like regression or panel with fixed effect estimation or the random effect estimation (see for instance, Baltagi (2008), Hsiao (2014) and Andre B et al. (2015)). Besides, the corresponding tests to determine which is the best method of estimation is needed (see the Hausman test, the Breusch and Pagan (1980)´s Lagrange multiplier, the F test for fixed effects to test whether all unobservable individual effects are zero); The authors talk about the relationship between these variables, however they do ot support the empirical evidence providing panel cointegration tests that are crucial (see for instance Kao (1999) panel data cointegration test, the Pedroni (1999, 2004) panel data cointegration test or the Westerlund (2005) panel data cointegration test, among others).

10. In the same time the authors must present the results of Dumitrescu-Hurlin test.

11. In the same time, the authors should test the validity of the model with the help of residual analysis using the Jarque-Berra test.

12. The conclusions at the end of the paper should be expanded showing the policy implications of the research results.

In conclusion, the article should be improve. It should also be enhanced with a review of the literature adequate to the subject and a broader interpretation and commentary of the research results.

6. PLOS authors have the option to publish the peer review history of their article (what does this mean? ). If published, this will include your full peer review and any attached files.

**Do you want your identity to be public for this peer review?** For information about this choice, including consent withdrawal, please see our Privacy Policy .

Reviewer #1: No

Reviewer #2: No

---

## [Author Response · Author response to Decision Letter 1]

19 Mar 2025

We would like to thank both reviewers for their positive evaluation of the manuscript and for their comments. In this letter we address each reviewer’s comment, and point to the changes we made in the revised manuscript according to their suggestions.

Reviewer #1

1. There were spots throughout the introduction where a citation was needed and there isn’t one. For example, the definition of social anxiety. Please add more citations throughout the introduction to back up claims.

Response: We have added an operational definition of social anxiety in the Introduction (page 5 of the revised manuscript, highlighted in yellow) and included new citations where appropriate (citations #19 and #20 in the revised manuscript). The new paragraph now reads: “Social anxiety can be defined as the tendency of individuals to be wary of others and to experience discomfort in social interactions. Low levels of social anxiety are normally experienced in social interactions, especially when new and/or unexpected. In a minority of the population affected by a social anxiety disorder, this experience of negative emotions is far more intense and accompanied by strong physiological arousal and leads to a significant impact on their daily functioning and well-being.”

2. Was there a directional hypothesis about the relation between social anxiety and stress reactivity in each group. Different possibilities were discussed but there did not appear to be a specific hypothesis.

Response: No, we did not have a strong hypothesis regarding the direction of the relationship between social anxiety and stress reactivity. We discussed in the Introduction both alternative possibilities, to disclose the exploratory nature of this specific part of our study (Page 7, Introduction section).

3. Were those in the stranger condition in romantic relationships? I understand that they were paired with strangers but did they have relationships outside of this study? If not, there might be systematic differences between the groups. There might be differences in how people perceive support from others based on their own relationship experiences so this might be listed as a limitation if they were not in relationships.

Response: Regarding the reviewer’s observation about relationship status within our sample, we acknowledge that only 50% of the participants in the Stranger group (N = 11) were involved in a romantic relationship during the study period, while the remaining participants were not. Following the reviewer’s valuable suggestion, we have explicitly incorporated this limitation in the conclusions section of our manuscript, which now reads: “A fourth limitation is represented by the fact that only half of the participants included in the Stranger condition were in a romantic relationship at the time of the research. At present, we can not rule out that being in a romantic relationship might have an impact even when facing a stressor together with a stranger providing support. In future studies, this feature should be controlled in order to evaluate possible differences in stress reactivity that could depend entirely on being or not in a romantic relationship, independent from having support when facing a stressor. ” (Page 20 of the revised manuscript, highlighted in yellow)

4. Was there rationale for always having the target participant be female? This was not set up in the introduction and came as a surprise

Response: We included only female participants to minimize the confounding effect of the known gender differences in emotional reactivity. We report this information in the conclusions highlighting that this poses a constraint on the generalizability of our findings: “This choice was made to minimize the potential confounding effects of the gender differences that exist in affective and psychophysiological reactivity, and in line with previous research showing that stress buffering on autonomic reactivity is larger in women. Nonetheless, this warns against a generalization of these effects also to the male population.” (Page 19 of the revised manuscript, highlighted in yellow)

5. What was the reliability of the STAI and SIAS scales in the current sample?

Response: We computed the Chronbach’s alpha for both measures and found excellent reliability. For SIAS the Chronbach’s alpha was 0.91, while for STAI it was 0.94. We reported this information in the revised manuscript (pages 12 and 14 of the revised manuscript, highlighted in yellow)

6. Did the two groups differ on any demographic variables? Baseline statistics should also be reported, perhaps in a table.

Response: The groups did not differ in terms of age. This information is now reported in the manuscript (page 8 of the revised manuscript, highlighted in yellow).

7. Were there any differences in stress reactivity or anxiety based on the gender of the supporter?

Response: We tested this possibility by refitting the model including also the gender of the supporter as a predictor and compared the original model with this second model with a likelihood-ratio test (LRT). The significance of the LRT would indicate that the additional variable (gender of the supporter) would increase the explained variance in the data. Results indicate that the gender of the supporter does not contribute to explaining neither stress reactivity nor anxiety. The reviewer can find the results of these analyses below:

LRT result for anxiety

Models:

mod_state_anxiety: stai_s ~ time * group + (1 | dyad)

mod_state_anxiety_2: stai_s ~ time * group * gender_alt + (1 | dyad)

npar AIC BIC logLik deviance Chisq Df Pr(>Chisq)

mod_state_anxiety 10 1179.4 1210.2 -579.70 1159.4

mod_state_anxiety_2 18 1187.0 1242.4 -575.53 1151.0 8.352 8 0.3999

LRT result for stress

mod_stress: stress ~ time * group + (1 | dyad)

mod_stress_2: stress ~ time * group * gender_alt + (1 | dyad)

npar AIC BIC logLik deviance Chisq Df Pr(>Chisq)

mod_stress 10 681.97 712.66 -330.98 661.97

mod_stress_2 18 694.20 749.44 -329.10 658.20 3.768 8 0.8774

8. I am struck by the difference in the physiological and psychological results. I agree with the authors’ proposal that the emotional concern might outweigh the safety signal of their partner or that performing in front of a partner is an added evaluative pressure. I have a potential third option as well. The TSST is a fairly negative experience and I’m wondering if it is possible that the target participant feels safer to express negative feelings in the presence of their significant other.

Response: The third option offered by the reviewer is very interesting. We included it in the conclusion of our revised manuscript as a potential alternative interpretation and suggested future experimental design that could help clarify this issue. The new paragraph now reads: “Future investigation should focus on designing more complex manipulation of social support. Indeed, a potential alternative interpretation for finding that participants performing in front of their partner report higher anxiety than those performing in front of a stranger, is that the former could have felt more comfortable in disclosing their levels of negative emotions due to their partner's presence. Explicitly manipulating the behavior of the partner could help clarify when its presence can act as a source of emotional buffer or as an additional source of psychological distress.” (page 20 of the revised manuscript, highlighted in yellow).

Reviewer #2

1. The Manuscript needs careful English proofreading because there are some shortcomings. For instance, the article “the” is sometimes missing in front of nouns, the message in some paragraphs is not clear enough. It looks like the first part was written by one author with a greater command of the English language, and the rest of the paper was written by someone else. The numerous grammar errors made this a difficult paper to read. It was strange to see the authors refer to tables that were not submitted. I was unable to find any supplementary material to the submission, so I think this was truly omitted by the authors. Please read the manuscript carefully.

3. The research is well based on science and the results are in agreement with the theoretical part. From my point of view, the paper is original and the topic addressed brings added value to the specialized literature regarding the anxiety. The paper is well written and easy to read.

Response: We sincerely appreciate the reviewer thoughtful feedback, particularly the kind words regarding the paper clarity and readability in comment #3. The positive remarks are truly encouraging. With regards to the suggested proofreading, we asked our university proofreading services and revised the manuscript accordingly. We would also like to address a potential misunderstanding concerning supplementary materials. We apologize if this caused any confusion, but we did not include any supplementary materials in our original submission. This explains why you were unable to locate them during your review. Thank you once again for your valuable input and the time you have dedicated to reviewing our work.

2. It would be very useful to add in the Introduction section the purpose, objectives and hypothesis of the research. I consider that a weak point of the paper is that the authors did not show the novelty of the paper compared to other works. That is why, I consider that the introduction should specify the novelty of the paper compared to other papers published in this area.

5. The Literature Review and the conclusions sections cannot be missing from the paper

Response: In the revised version of the manuscript, we have now better distinguished the hypotheses and the goals of the manuscript, adding a new subsection in the Introduction section titled “Research objectives” (page 6 in the revised manuscript, highlighted in yellow). Following comment #5, we have now separated the discussion from the conclusions sections of our manuscript to improve clarity according to the reviewer’s suggestion.

6. In my opinion, the results from the Statistical Analysis section, which presents three models with dependent variables and random effects, should be presented in a table. Here, the authors should apply the Hausman and VIF tests (see: https://doi.org/10.1371/journal.pone.0283277).

7. Authors must refer to the software used for econometric analyses: STATA, EViews, SPSS etc.

8. The authors must also show the values of the adjusted R-square, which is more relevant in the methods used in this paper.

Response: We reported in our revised manuscript that all the statistical analyses were performed using R 4.1.2 (page 11 in the revised manuscript, highlighted in yellow). Following the reviewer’s suggestion, we reported the results for each model in tables (Table 1, Table 2 and Table 3) and reported the additional information requested by the reviewer (R2, VIF). With regards to VIF for all three regression models, VIF was 1, suggesting no collinearity.

4. The research is well based on science and the results are in agreement with the theoretical part. The model applied to the analyzed data is correctly used in the analysis undertaken, it is a strength point of this paper.

9. The authors applyed the regression method, then they must apply the following tests: the authors must to apply an econometric method like regression or panel with fixed effect estimation or the random effect estimation (see for instance, Baltagi (2008), Hsiao (2014) and Andre B et al. (2015)). Besides, the corresponding tests to determine which is the best method of estimation is needed (see the Hausman test, the Breusch and Pagan (1980)´s Lagrange multiplier, the F test for fixed effects to test whether all unobservable individual effects are zero); The authors talk about the relationship between these variables, however they do ot support the empirical evidence providing panel cointegration tests that are crucial (see for instance Kao (1999) panel data cointegration test, the Pedroni (1999, 2004) panel data cointegration test or the Westerlund (2005) panel data cointegration test, among others).

10. In the same time the authors must present the results of Dumitrescu-Hurlin test.

11. In the same time, the authors should test the validity of the model with the help of residual analysis using the Jarque-Berra test.

Response: We appreciate the Reviewer's positive feedback on our statistical approach in comment #4, where they found our analyses appropriate for the study’s goal and highlighted them as a strength of the paper.

Regarding the suggested methods in comments 9-11 (Hausman test, panel cointegration test, Lagrange multiplier test, Dumitrescu-Hurlin test, and Jarque-Berra test), we acknowledge that these econometric methods have an established value in financial and economic research contexts. However, these methods are not commonly employed in experimental psychology and psychophysiological research. We have opted to maintain our current statistical framework to ensure alignment with methodological standards in our field, thereby preserving readability and potential impact for our target audience. We believe this approach best serves the paper's objectives while adhering to rigorous methodological standards.

---

## [Decision Letter · Decision Letter 1]

12 May 2025

Social anxiety influences the stress-buffering potential of social presence: Evidence from cardiovascular and affective reactivity under stress

PONE-D-24-26805R1

Dear Dr. Maffei,

We’re pleased to inform you that your manuscript has been judged scientifically suitable for publication and will be formally accepted for publication once it meets all outstanding technical requirements.

Kind regards,

Annalisa Theodorou

Academic Editor

PLOS ONE

Additional Editor Comments (optional):

Both the reviewers and I think that the authors sufficiently addressed all the comments, and the manuscript deserves to be published as it is. Congratulations to the authors for this very interesting study and the publication of their work.

Reviewers' comments:

Reviewer's Responses to Questions

**Comments to the Author**

1. If the authors have adequately addressed your comments raised in a previous round of review and you feel that this manuscript is now acceptable for publication, you may indicate that here to bypass the “Comments to the Author” section, enter your conflict of interest statement in the “Confidential to Editor” section, and submit your "Accept" recommendation.

Reviewer #1: All comments have been addressed

Reviewer #2: All comments have been addressed

2. Is the manuscript technically sound, and do the data support the conclusions?

Reviewer #1: (No Response)

Reviewer #2: Yes

3. Has the statistical analysis been performed appropriately and rigorously? 

Reviewer #1: (No Response)

Reviewer #2: I Don't Know

4. Have the authors made all data underlying the findings in their manuscript fully available?

Reviewer #1: (No Response)

Reviewer #2: Yes

5. Is the manuscript presented in an intelligible fashion and written in standard English?

Reviewer #1: (No Response)

Reviewer #2: Yes

6. Review Comments to the Author

Reviewer #1: (No Response)

Reviewer #2: All comments have been successfully addressed, and the authors have provided comprehensive responses that strengthen the rigor of their analysis. Key methodological considerations have been effectively incorporated and clarified. Furthermore, robustness checks and supplementary analyses have been conducted where necessary, reinforcing the validity of the study's findings.

Given these improvements, the manuscript is well-prepared for publication. The authors have demonstrated a thorough understanding of the relevant literature and policy implications, ensuring that the study makes a valuable contribution to the field. The revisions provide clarity and depth, enhancing the overall quality of the work.

7. PLOS authors have the option to publish the peer review history of their article (what does this mean? ). If published, this will include your full peer review and any attached files.

**Do you want your identity to be public for this peer review?** For information about this choice, including consent withdrawal, please see our Privacy Policy .

Reviewer #1: No

Reviewer #2: No

---

## [Editor Report · Acceptance letter]

PONE-D-24-26805R1

PLOS ONE

Dear Dr. Maffei,

I'm pleased to inform you that your manuscript has been deemed suitable for publication in PLOS ONE. Congratulations! Your manuscript is now being handed over to our production team.

Kind regards,

on behalf of

Dr. Annalisa Theodorou

Academic Editor

PLOS ONE